# The Use of Neuromodulation for Symptom Management

**DOI:** 10.3390/brainsci9090232

**Published:** 2019-09-12

**Authors:** Sarah Marie Farrell, Alexander Green, Tipu Aziz

**Affiliations:** 1Nuffield Department of Surgical Sciences, John Radcliffe Hospital, University of Oxford, Oxford OX3 9DU, UK; sarah.farrell@medsci.ox.ac.uk; 2Nuffield department of clinical Neurosciences, John Radcliffe Hospital, University of Oxford, Oxford OX3 9DU, UK; alex.green@nds.ox.ac.uk

**Keywords:** neuromodulation, deep brain stimulation (DBS), pain, dyspnoea, blood pressure, hypertension, orthostatic hypotension, micturition, bladder control

## Abstract

Pain and other symptoms of autonomic dysregulation such as hypertension, dyspnoea and bladder instability can lead to intractable suffering. Incorporation of neuromodulation into symptom management, including palliative care treatment protocols, is becoming a viable option scientifically, ethically, and economically in order to relieve suffering. It provides further opportunity for symptom control that cannot otherwise be provided by pharmacology and other conventional methods.

## 1. Introduction

Symptom management is an opportunity to alleviate suffering, whether or not a disease state is curative. This can range from an able-bodied young individual with intractable cluster headaches, to an elderly patient with non-curative cancer whose main hindrance to quality of life in their last months or years is breathlessness. The realms of palliative care extend outside that of end-of-life to encompass those living with intractable suffering. The Centre to Advance Palliative Care defines Palliative medicine as “specialized medical care for people living with serious illness” [1]. This suffering often entails symptoms of chronic pain and a variety of dysautonomias including dyspnoea, micturition dysfunction, and cardiovascular problems. For those suffering despite their current medication regime, the notion of a life free from constant pain, relief from the sensation of breathlessness, with the ability to control bladder issues, seems nothing short of a miracle.

Neuromodulation (deep brain stimulation, motor cortex stimulation, spinal cord stimulation, dorsal root ganglion stimulation) is a safe and effective treatment, largely deployed for movement disorders including Parkinson’s disease tremor and dystonia [2,3], as well as epilepsy [4], psychiatric disorders such as depression/obsessive compulsive disorder/Tourette’s [5,6], and a variety of previously intractable chronic pain syndromes [7,8]. Through the serendipitous amelioration of autonomic problems in patients previously fitted with these electrodes, there is a growing body of evidence demonstrating the ability of neuromodulation to ameliorate adverse autonomic effects associated with breathlessness, micturition, and cardiovascular function. These findings have led to further investigation surrounding neuromodulation and autonomic function, with kind involvement from patients previously fitted with these devices. Gaining control of these dysautonomias could palliate thousands of patients who are suffering, whether this be end-of-life or otherwise.

The brain works via a complex flow of signal processing and, when wiring is faulty, can prove difficult to correct with current mainstream pharmacology. Persuading these suboptimal networks to behave optimally is the realm of ‘neuromodulation’. For the most part, this means electrodes are implanted into the brain or spine and the signals optimized to achieve a particular response, either by effectively ablating the area, or stimulating/exciting the area, though the exact mechanism of action is equivocal. The use of neuromodulation is attractive because of its reversibility, its targeted localised delivery, and the ability to adjust settings to optimize the effects (frequency, amplitude, pulse width of current delivered).

For certain conditions such as Parkinson’s disease, dystonia, and tremor, neuromodulation with medical management is now established therapies and is managed by specialist neurological centres. This discussion will centre around indications for which there are little or no non-surgical alternatives. The article moves through the evidence pertaining to neuromodulation and its ability to ameliorate the symptoms of pain, hypertension, orthostatic hypotension, dyspnea, and micturition, considering the economic and ethical aspects of this care. It demonstrates the promise of neuromodulating symptoms previously intractable to pharmacology and more conventional surgery.

### Methodology

A PubMed search of literature was conducted describing ‘DBS’ (deep brain stimulation), ‘neuromodulation’, or ‘spinal cord stimulation’ with the following search terms: ‘pain’ and ‘autonomic function’, ‘cardiovascular’, ‘blood pressure’, ‘heart rate’, ‘micturition’, ‘bladder’, ‘respiratory’, and/or ‘breathing’. All references found were scanned for relevance, categorized by intractable disease type, and then reviewed in more detail. The relevant references found in these articles were also added to the list.

## 2. Pain

A significant number of patients suffer from intractable pain, as much as 29% of the adult population in Europe have moderate-to-severe pain [9], as well as 100 million people in the Unites States [10,11]. This carries emotional and cognitive sequelae for those it affects [12,13]. Moreover, the opioid epidemic compounds and conflates this issue. Alternatives are sought. Neuromodulation for pain is well established. Over the past few decades, the periaqueductal grey (PAG), thalamus, and more recently the anterior cingulate cortex (ACC) have become popular targets. Meanwhile, the rise of spinal cord stimulation (SCS) and dorsal root ganglion stimulation has provided a viable option.

A comprehensive review of pain and neuromodulation can be found elsewhere [7], aside from the topic of cluster headaches, which can be found below. In summary, neuromodulation is already a well-established treatment for pain, with many success stories. Outcomes are generally favourable for SCS, particularly with newer generations of technology such as burst stimulation and dorsal root ganglion implants. Studies of DBS (targeting PAG and ACC) outcomes tend to be more heterogeneous, though the Oxford group have found this treatment to be beneficial for many patients of varying pain aetiologies [14]. It might be reasonable to think that types of chronic pain, ‘nociceptive versus deafferentation’ or ‘central versus peripheral’, could be categorized as more or less amenable to neuromodulation. However, the involvement of neuronal plasticity encompasses centrally mediated changes, as shown by functional neurology and electrophysiological studies [15,16,17]. It may be better to select those eligible for neuromodulation by an absence of psychogenic elements, which would otherwise rule out the ability of neuromodulation to help [7]. There is a multitude of issues plaguing chronic pain trials that may lead to less-than-favourable results, including the nonrandomised nature, heterogeneity of patient aetiology, subjective unblinded assessment of patient outcome, inconsistent stimulation parameters, sites and number of electrodes implanted, the use of ‘50% reduction in pain’ as a set threshold, and the treatment often taking place only when SCS has been unsuccessful. Thus, there are many patients for whom DBS has reduced pain and improved quality of life, but are represented as a ‘fail’ in the literature. The modest literature results reflect the inability of the data to represent the potential for DBS. Success rates vary between 30% and 100% pain relief in certain cases (pending long term follow up required). While it is true that published studies demonstrate an average 30% relief [18], this is no small fete for those individuals with a 30% (or more) relief in pain. Fundamentally, and of particular importance when discussing palliative treatment, for an individual with intractable pain who has already tried the limited treatment options available to them, a success rate lower than an arbitrary 50% cut off point versus a 1:500 risk of stroke from surgery may be a reasonable risk for this individual to take. Of course, the results vary depending on indications, follow-up times, dedication to optimisation of settings, and so on. Patient selection is vital, presurgical neuropsychological evaluation is required to restrict surgery to those without risk factors for negative outcomes including catastrophization, opiate addiction, low activity levels, and even ongoing litigation [19].

### Posterior Hypothalamus as a DBS Target for Cluster Headaches

DBS for cluster headaches has already experienced some success: this debilitating syndrome is known to have key elements of parasympathetic activity producing the classic ipsilateral symptoms (conjunctival injection, Horner syndrome, and lacrimation) in conjunction with intense pain [20]. The hypothalamus is thought to be the source of this autonomic dysfunction on the basis of activations demonstrated in positron emission tomography (PET) and functional magnetic resonance imaging (fMRI) studies, as well as the regularity of the daily attacks in keeping with the role of the hypothalamus in circadian rhythm [21]. Stimulation of the posterior hypothalamus has been shown to increase sympathetic activity [22], and thus is utilised as a treatment for refractory cluster headache [23,24,25]. Success rates from continuous stimulation have been hopeful, with one study showing that 13/16 patients experienced pain abolition or major pain reduction two years after surgery, though declining to 10 patients at the four-year follow up [26]. In a prospective, double-blind crossover study in France, 11 patients were randomized to either sham of active surgery for a month followed by a one-year open phase, using weekly attack frequency as the primary outcome. Although no difference was observed between groups in the randomized phase, the one-year open phase indicated long-term efficacy in over 50% of patients without high morbidity [27].

## 3. Blood Pressure Control

Neuromodulation can alter cardiovascular parameters, with blood pressure (BP) being of particular interest. This has the potential to manipulate refractory hypertension (HTN) and orthostatic hypotension. The ability of the midbrain to modulate blood pressure was described in the cat in 1935 [28]. Through intraoperative investigations on the human brain, we see the predominant area of interest is the PAG, though there are intriguing findings in the posterior hypothalamic area (PHA) and the subthalamic nucleus (STN) Table 1 summarises key studies involving DBS and blood pressure.

### 3.1. Refractory Hypertension and Deep Brain Stimulation

Thirty-three percent of the U.S. population are affected by HTN [29], one of the greatest risk factors for cardiovascular disease. Less than half receive appropriate BP control and 0.5% of these are refractory medical treatment [30,31]. The definition of refractory HTN involves trying at least five different medications without fully rectifying BP problems, including a mineralocorticoid receptor antagonist and a thiazide-like diuretic [32,33]. It is thought to be the result of sympathetic outflow (whereas other types of HTN may be predominantly owing to hypervolaemia). Ameliorating refractory HTN is an important issue. Patients with ‘resistant’ HTN (elevated BP despite three or more medications) have higher risk of cardiovascular issues including stroke, myocardial infarction, aneurysm formation, heart failure, end stage renal disease, cardiac arrhythmia, hypertensive encephalopathy, hypertensive retinopathy, glomerulosclerosis, limb loss due to arterial occlusion, and sudden death [33]. While resistant HTN is better described, cardiovascular outcomes from refractory HTN indicate that these risks are further increased [33].

### 3.2. PAG as a DBS Target for Hypertension

PAG stimulation is known to modulate cardiovascular parameters. Animal studies show electrical stimulation of the ventrolateral columns of the PAG lowers HR and BP, as well as producing freezing behaviour. In contrast, stimulation of the dorsolateral and dorsomedial produces increased heart rate and BP [35,44,45,46]. The separation of the PAG into four functional columns could potentially be used for differing palliative treatments. Ventrolateral stimulation would be desirable for HTN, whereas stimulation of dorsomedial and dorsolateral columns would prove useful in cases of orthostatic hypotension [38,43].

DBS PAG has produced evidence of changes in blood pressure. Green et al. found stimulation of the ventral PAG caused a 14.2 mmHg ± 3.6 mean reduction (from 143 to 128 mmHg) in systolic and 4.9 mmHg reduction in diastolic BP [47]. In line with this, one hypertensive patient—with a previously fitted PAG stimulation for chronic pain—experienced their baseline arterial pressure fall from 157.4/87.6 mmHg to 132.4/79.2 mmHg [41]. More recently, O’Callaghan et al. reported a patient who had previously tried a variety of medications to control her blood pressure, including a baroreceptor activation device. After six months of ventral PAG stimulation, her average morning blood pressure dropped from a pre-surgery value of 280/166 mmHg to 210/130 mmHg [48].

Similar blood pressure reductions can be sustained one year after surgery [36,49,50]. While it is possible these reductions may be confounded by analgesic benefit, one study demonstrated a separation between the two. Left hemibody pain (under PAG stimulation) returned to baseline 4 months after surgery, but at 27 months, when DBS stimulation was turned off, blood pressure rose by 18.5 mmHg, indicating that there had been a continued effect separate from pain relief [17].

### 3.3. PAG as a DBS Target for Orthostatic Hypotension

The original Green et al. study from 2005, demonstrating a decrease in BP upon ventral stimulation, also showed dorsal stimulation to have the opposite effect, that is, mean increase in systolic BP of 16.7 ± 5.9 mmHg (*n* = 6) [47]. A further study by the same group explored whether orthostatic hypotension could be treated with PAG stimulation. Eleven patients previously implanted for chronic pain were divided into three groups—those with orthostatic hypotension (*n* = 1), those with mild orthostatic intolerance (*n* = 5), or those with no orthostatic intolerance (*n* = 5). When ‘on’ stimulation, the decrease in systolic BP while moving from sitting to standing was reduced from 28.2% to 11.1% in the patient with orthostatic hypotension [51]. For those with mild orthostatic intolerance, the systolic BP (initially 15.4% drop) was completely reversed. The remaining group experienced no side effects. This amelioration of orthostatic hypotension was found without increasing baseline resting BP, but with an associated increased heart rate response to standing.

The effects of PAG stimulation on the cardiovascular system may be understood as there is evidence that PAG projects to preganglionic cardiac vagal neurons in the nucleus ambiguous, with chemical stimulation of PAG in rats inhibiting baroreflex vagal bradycardia [34].

Overall, ventral PAG stimulation seems beneficial for HTN, whereas dorsal PAG stimulation may improve orthostatic hypotension. Given that orthostatic hypotension is present in 30% of adults 70 years and older [39], contributing to increased fall risk, cardiovascular disease, stroke, and death [37], the ability to mitigate these BP difficulties is clinically significant. We have yet to purposefully target ventral versus dorsal columns, though there is no reason this could not be attempted in future. Given this convincing evidence of the effect of PAG on BP, it is surprising to note there are no clinical trials registered at clinicaltrials.gov (using search terms of ‘DBS’, ‘neuromodulation’, ‘deep brain stimulation’ and ‘blood pressure’, ‘hypertension’, or ‘orthostatic hypotension’). This may speak to the low number of PAG stimulated chronic pain patients available for the research.

### 3.4. Subthalamic Nucleus as a DBS Target for Orthostatic Hypotension 

Although two studies of head up tilt table testing (HUTT) in patients undergoing DBS of the STN reported no beneficial results [42,52], several studies demonstrate an increase in, or maintenance of, both arterial BP and baroreceptor sensitivity [53,54,55]. Stemper et al. demonstrated that patients with STN DBS were able to maintain BP and baroreflex sensitivity, without which they experienced significant orthostatic hypotension during HUTT [53]. As neither BP or baroreflex sensitivity were influenced by stimulation of motor thalamus, globus pallidus interna (GPi), pedunculopontine nucleus (PPN), or ACC, this suggests some anatomic specificity to STN. In keeping with this, Hyam et al. demonstrated that PD patients who are stimulated in the STN with a frequency over 100 Hz experience a modest increase in HR and BP (5 bpm and 5 mmHg, respectively) [36]. Sverrisdottir et al. also demonstrated increase in orthostatic tolerance in Parkinson’s disease (PD) patient with STN DBS [54].

Although some studies show this improvement in orthostatic hypotension is sympathetically-mediated [40,56,57], Sumi et al. highlighted the possibility that any cardiovascular improvements seen with STN DBS could be the result of the increased ability to exercise rather than the stimulation itself [58], or a reduced pharmacological requirement of the Parkinson’s disease state [58]. However, peri-operative studies showing cardiovascular changes suggest there may be a more direct role of STN. Hyam points out this may be because of the high frequencies of STN leading to exposure of stimulus to nearby areas of the central autonomic networks [36]. This is supported by earlier findings that showed the one STN patient (out of the five) who experienced autonomic alterations was the same patient that had a lead placement extending to posteromedial and lateral hypothalamic areas [22].

### 3.5. Posterior Hypothalamus as a DBS Target to Ameliorate Orthostatic Hypotension

With the advent of using DBS for PD, multiple patients have been tried and tested, allowing variation across electrode placement positions. Electrodes inadvertently placed close to posterior hypothalamus showed that BP and respiratory rate increased after stimulation [22]. It would seem that stimulation of posterior hypothalamus may facilitate the maintenance of SBP during head up-tilt testing. As previously mentioned, patients with cluster headaches can have stimulation in the posterior hypothalamic area (HPA). When off stimulation, HUTT resulted in a 3% fall in systolic blood pressure, whereas systolic BP was maintained when the test was repeated during stimulation [22]. 

## 4. Dyspnoea in Chronic Obstructive Airway Disease

Dyspnoea is difficult to treat and distressing for the patient. It is the most important complaint in many common respiratory diseases such as chronic obstructive airway disease (COPD) [59]. COPD affects 6% of the population and is a major cause of morbidity and mortality worldwide [60]. Many other diseases lead to this distressing sensation of breathlessness including asthma, lung cancer, and end-stage heart failure, to name a few. Neuromodulation may be able to tap into the central control signals and provide some relief. Bronchoconstriction causes increased airway resistance and is a key element of the pathology behind asthma and COPD. Airway smooth muscle is mediated by airway-related vagal pre-ganglionic cells (AVPN)—part of the parasympathetic nervous system. Relaxing these airways requires the circulating catecholamines of the sympathetic nervous system. It is, therefore, feasible to control the lungs via the brain.

Since 1894, the effect of orbitofrontal cortex stimulation on respiratory rate was demonstrated in a variety of animals including cats, dogs, and monkeys [61]. Manipulation of the respiratory rate was later shown by stimulation of the anterior cingulate cortex during open brain surgery [62].

Hyam et al. studied the effect of stimulation on peak expiratory flow rate (PEFR) and forced expiratory volume in one second (FEV1) in 17 chronic pain patients and 20 movement disorder patients. Within the pain syndrome patients, 10 had stimulation in the PAG (an area with connections to AVPN) and 7 in the sensory thalamus. A similar control was used for the movement disorders, with 10 STN stimulated patients (relevant to the AVPN) and 10 GPi stimulation patients. Spirometry recordings were made and the PEFR was taken, along with the FEV1. PEFR is defined as the highest flow achieved from a maximum forced expiratory manoeuvre started without hesitation from a position of maximal lung inflation. FEV1 is defined as the maximal volume of air exhaled in the first second of a forced expiration from a position of full inspiration. The experimenters also recorded the maximal volume of air exhaled with maximally forced effort from maximal inspiration, or forced vital capacity (FVC).

STN and PAG both improved PEFR, but not FEV1. Mean PEFR percentage change was 13.4 ± 4.6% with PAG stimulation, a non-significant 0.89 ± 2.6% with sensory thalamus stimulation, 14.5 ± 5.3% with STN stimulation, and −0.2 ± 1.8% with GPi stimulation [63]. The lack of significant change in FEV1 could be related to either the STN and PAG affecting upper airways more, or the ability of airway flow changes in FEV1 to be seen only if the patients suffered from respiratory disease. This is bolstered by the one patient who showed an obstructive lung function profile on spirometry—exhibiting a 9.8% increase in FEV1 on PAG stimulation. This effect of stimulation is comparable to the effects of nebulised or oral steroid use seen in acute exacerbation of COPD [64,65,66] and lacks the risk of toxic side effects. Given that patients stimulated in the sensory thalamus and GPi showed no change in lung function, we can assume these changes are not related to the decrease in pain or amelioration of motor disorder.

It is reasonable to suppose that STN stimulation improves lung function, having previously been implicated in the respiratory network. STN is expected to be active during breath holding and has a role in inhibiting initiated responses in stop-signal paradigms [67,68]. The PAG is also recognised to be integral in the freeze–flight phenomenon, hence stimulation of PAG causes changes in the cardiovascular system, vocalization, and micturition [69,70,71]. It is logical that these sympathetically innervated actions would also extend to increasing respiratory function, preparing the body to fight or flee. The PAG projects to the parabrachial nuclei and stimulation of the nuclei in animals have been implicated in cardiorespiratory variables: lesions in this region can cause distortions of the Hering–Breuer reflex [72,73], whereas chemical stimulation in anaesthetized cats causes reduced total lung resistance [74,75].

The management of dyspnoea in patients with chronic airway diseases requires more attention; currently, the American Thoracic Society recommends bronchodilators as the mainstay of treatment, with lung volume reduction surgery and anxiolytic therapies considered on an individual basis [76]. It would be extremely interesting to see the benefit of DBS in COPD, asthma, and sleep apnoea patients. To our knowledge, there is currently only one clinical study registered on clinicaltrials.gov related to DBS and respiratory function. This study, among other things, will look at how ‘on’ versus ‘off’ stimulation affects patients with multisystem atrophy who tend to experience decreased respiratory function. See Table 2 for clinical trials relating to pain and dysautonomias and Table 3 for key studies surrounding DBS and dyspnoea.

## 5. Micturition

Lower urinary dysfunction (LUTD) is disabling [77], difficult to treat, and extremely common in neurological diseases such as PD [78,79,80] and multiple sclerosis [81]. In fact, these symptoms occur in 74% of patients with PD, with severe symptoms in over 50% [82].

LUTD is an increasing common symptom worldwide referring to urgency, increased urinary frequency, or incontinence, with neurological disorder being one of the main causes. Urodynamic examination in these patients shows detrusor hyperreflexia with involuntary contractions of the bladder, resulting in a reduced bladder capacity and an early desire to void. Sacral neuromodulation (SNM) is established as a treatment for non-neurogenic LUTD, but a more centralised method may be required to achieve results for neurogenic LUTD given the lack of widespread success of SNM for LUTD [83]. It is feasible that central neuromodulations are a potential treatment approach given the complex network of autonomic and central nervous system control of micturition reflexes [84,85]. The pontine micturition centre (PMC) is the central micturition reflex centre for the brain [86,87]. Pathways of the lower urinary tract system consist of an afferent pathway from the urinary bladder to the PMC via the pelvic nerve and spinal cord, and an efferent pathway projecting from the PMC to the bladder via the sacral parasympathetic centre of intermediolateral column cells [88]. The PAG, locus coeruleus (LC), and rostral pontine reticular nucleus are all thought to modulate bladder activity via their connections to the PMC.

We can think of the bladder as being in a ‘storage mode’ and switching to voiding when socially appropriate. Research looking to optimise urinary function using DBS stems from movement disorder and chronic pain patients willing to undergo urodynamic testing.

### 5.1. STN and Micturition

Both animal and human studies show improved urodynamics with STN stimulation, measured by bladder capacity and first desire to void [89,90,91,92]. For example, Seif et al. report initial desire to void at 199 ± 57 mL, with maximal bladder capacity at 302 ± 101 mL. This contrasts with the ‘off’ stimulation condition (initial desire to void occurring at mean value of 135 ± 43 mL). The maximal capacity of the bladder was 174 ± 52 mL, with both values shown to be significantly different in the on and off stimulation (*p* < 0.005) [92]. Thus, DBS of STN reduces detrusor overactivity and increases bladder capacity, effectively normalising the ‘storage’ phase.

Positron emission tomography (PET) studies show that ‘off’ stimulation bladder-filling in PD patients with bilateral STN stimulation increases regional cerebral blood flow to ACC and lateral frontal cortex [91]. This suggests that ‘on’ stimulation may ameliorate the bladder dysfunction by effective integration of afferent bladder information. Future studies should address the effects of STN DBS in patients specifically with continence problems.

### 5.2. Thalamus and Micturition

Studies of thalamus and micturition suggest a negative effect on lower urinary tract symptoms. Thalamic DBS has been shown to significantly decrease bladder volume at ‘first’ and ‘strong’ desire to void, as well as decreasing maximal bladder capacity [93,94]. This demonstrates that although the thalamus has a role in micturition, it is not a target for rectifying dysfunction.

### 5.3. PAG and Micturition

The PAG has previously been hypothesized to be a micturition switch, changing bladder state from ‘storage’ to ‘voiding’. It is responsible for sensory inputs from distended bladder-activated spinal–midbrain–spinal nerve circuitry. Lumbrosacral neurons are known to terminate on neurons in the PAG [95], and the PAG then projects to the PMC [96]. PAG neurons are activated during voiding [97], and animal studies have previously shown that PAG elicits micturition effects—both stimulatory [95,98] and inhibitory [99].

In humans, stimulation in PAG increases maximal bladder capacity, as judged by the volume at which patients, fitted with a catheter, ask for a saline infusion to be stopped. Increased subjective bladder capacity was found in the ‘on’ versus ‘off’ state (*p* = 0.028), though this did not affect volumes at which voiding was desired [93]. Of note, when ‘on’ stimulation while the bladder was filling from empty, the volume at which urine first started to escape from the penis (maximum cystometric capacity (MCC)) was greater compared with the ‘off’ stimulation. Interestingly, stimulation affected micturition over a much wider area of the PAG than the expected caudal ventrolateral part [100]. The Oxford group hypothesizes that rostrally located electrode placements in the PAG are most likely to activate afferent inputs to the caudal ventrolateral PAG [101], whereas more caudal stimulation activates local intrinsic connections to the ventrolateral PAG [100] or possibly descending efferent pathways [102,103]. It is also possible that the afferent pathway has antidromic activity from the cord, serving to cancel orthodromic afferent signals from the bladder. Activation of any of these circuits by DBS could potentially disrupt the micturition control network in the PAG either via the GABAergic synapses or by creating a non-physiological network activity pattern, blocking initiation of a synchronized voiding pattern in the detrusor and sphincter muscles. Thus, urine output is prevented even when the bladder is distended [97].

### 5.4. PPN and Micturition

The pedunculopontine nucleus (PPN) is a relatively new experimental brain target for managing severe Parkinson’s disease, and through studies relating to this, its involvement in micturition is highlighted. PPN DBS is known to improve akinesia and gait difficulties in both animals and patients with PD [104,105,106,107,108]. In 2011, Aviles-Olmos et al. described their findings of this stimulation causing detrusor over-activity immediately after right-sided PPN DBS [104]. This is supported by a study of Gottingen minipigs, where stimulation of the PMC resulted in increased detrusor pressure [109], measured by cystometry, and is plausibly because of the involvement of the pontine micturition centres and their connections.

Despite this, the Oxford group did not find detrusor overactivity of lower sensory threshold during bladder filling. They studied five patients with bilateral PPN for PD. In fact, stimulation provided significant increase in maximal bladder capacity averaging at 199 mL during the ‘on’ stimulation (range 103–440 mL) compared with 131 mL during the ‘off’ stimulation (range 39–230 mL) [110]. It is worth noting the considerable spread of response to stimulation in terms of bladder capacity, across subjects. Further investigation is required to attribute this to any particular cause, given that it was not linked to stimulation type (monopolar vs. bipolar), electrode location, or duration of stimulation. Interestingly, white matter tractography did not show either modulation of activity within the PAG (a proposed ‘micturition switch’) or involvement of many established bladder network components (e.g., insula cortex and ACC). An improved understanding of what is causing these varied and differing results are required before trials of this implant are made purely for micturition issues alone. 

In summary, the growing literature of DBS and micturition provides potential targets. Basal ganglia and brainstem targets (STN and PAG) inhibit micturition and improve incontinence. The results from the PPN also appear promising. However, thalamic targets induce micturition [81]. A 2017 DBS study in rats was the first study to evaluate the effects of conducting DBS on four potential targets on bladder activities. It suggests the Pedunculopontine tegmental nucleus is the most promising DBS target for developing new approaches to treat bladder dysfunction, being most efficient in suppressing reflexive isovolumetric bladder contractions compared with PAG, rostral pontine reticular nucleus, and locus coeruleus [111].

The rapid onset and reversibility of DBS allows the prospect of intermittent, patient-controlled use with minimal risk of side effects. This may be of particular use in those urinary incontinence syndromes of central origin such as Parkinson’s disease or poststroke incontinence. Current clinical trials pertaining to DBS and lower urinary tract symptoms are listed in Table 2. Table 4 lists key studies surrounding DBS and Micturition.

### 5.5. The Potential of DBS for Other Intractable Symptoms

Neuromodulation may also prove useful for other debilitating conditions proving intractable and carrying substantial morbidity. For example, sudomotor dysfunction is extremely common in PD patients and can make for uncomfortable sleep and awkward social functioning. One study showed that 34 out of 35 patients were completely relieved of the drenching sweats they experienced prior to STN stimulation [112] and several others demonstrate similar effects [77,113,114]. In contrast, electrodes mistakenly placed in thalamus or posterolateral hypothalamus have caused hyperhidrosis [22,115]. 

Additionally, there is evidence for the improvement of GI dysmotility with STN stimulation, including improved deglutition and faster pharyngeal transit times, leading to a reduced rate of aspiration during swallowing [116,117,118]. Improved gastric emptying has been demonstrated by 13C-acetate breath testing [40,119] and improved GI motility [40]. The latter study calculated frequency reductions of 50%–25% for dysphagia, 35–15% for sialorrhea, 95–75% for constipation, and 85–50% for problems with defecation. 

## 6. Cost Effectiveness of Neuromodulation for Intractable Suffering

There is no doubt that the list of symptoms discussed spans millions of people. Chronic pain alone affects approximately 100 million people in the United States, costing 635 billion dollars each year in medical treatment and lost productivity [11]. Hypertension effects 32.6% of the U.S. population [29], 5% of which is refractory to current medications [20,31]. Dyspnoea spans a whole host of respiratory diseases including COPD, heart failure, lung cancer, fibrosis, and asthma. Urinary urgency, frequency, nocturia, and incontinence are common symptoms in many neurological diseases including multiple sclerosis and Parkinson’s disease. 

The economic sequelae of palliating these symptoms in so many patients are difficult to calculate. Essentially, it is easy to see the costs, and hard to quantify benefits. There is an initial high cost of implants and follow up clinics. A systematic review of DBS for PD calculated that over five years, the cost of DBS for one patient would be 186,244 USD (amalgamated from nine studies) [120], and it would be reasonable to assume a similar value for other uses of DBS.

It is more difficult to weigh up the economic gains. These include increased productivity for those able to re-enter the workforce, less time waiting for clinic appointments or drug trials, reduced requirement of pharmacology therapy, reduced need for carers once symptoms have ameliorated, and patients can resume activities of daily living, and so on. It is worth noting that the economics of these changes depend on whether a patient is nearing end-of-life, as does their suitability on a fitness-for-surgery basis. The expansion of palliative care outside that of end-of-life shifts the balance towards neuromodulation.

## 7. Ethical Considerations for Symptom Management

While reduction in suffering is always desirable, deriving the extent to which finite resources should be devoted to this goal is more complex. A second layer of complexity results from the need to evaluate the extent to which additional risk should be incurred attempting interventions that may not be successful. This is particularly difficult in a palliative setting [121]. 

The ethics of providing neuromodulation is a complex formula calculated on a case-by-case basis including a multitude of factors: the general medical condition of the patient, effectiveness of the therapy in question, comparative effectiveness of other therapies available, amount of suffering encountered by the patient, life expectancy, and of course the cost of the therapy as it relates to the ability to provide for others with the resources available. On the latter point, it is hoped that neuromodulation can avoid increasing symptom management costs compared with current best practice, for example, by avoiding hospitalizations late in disease course. Parallels may be drawn here with chemotherapy costs in pain management, where research demonstrates that, compared with less expensive treatments, initial costs are levelled out with reductions in the care required further down the line [122].

## 8. Conclusions

There are millions of patients suffering from intractable debilitating symptoms whose treatment needs are not currently being met by conventional medications. Neuromodulation is a promising palliative method. Evidence weighs in heavy on the positive effect of neuromodulation for intractable pain, particularly through stimulating the PAG and ACC. This comes with the caveat of treating the right patient, in the right brain location, and with the right stimulation parameters. The patient must work with neurosurgeons and the rest of the pain team to find their ‘Goldilocks’ parameter settings.

Increasingly, research demonstrates a role for neuromodulation of autonomic dysfunction in a variety of settings ranging from blood pressure control, to micturition, to breathlessness. Firstly, it is theoretically possible that both uncontrolled hypertension and postural hypotension may be amenable to DBS in the future. Particular locations of interest include the PAG for HTN and STN for orthostatic hypotension. This has already been demonstrated in a small number of patients, both short and longer term. Secondly, evidence is accumulating for the control of micturition, which may help those suffering from detrusor instability and early desire to void. Future research can focus on the subgroup of patients (e.g., Parkinson’s patients) suffering from this and already receiving neuromodulation for motor problems. Current trials are listed in Table 2. Locations of interest include STN and PPN, with some unresolved hints that PAG may be of interest. Thirdly, while many potential patients could benefit from neuromodulation for dyspnoea, the research is in its infancy, but highlights the importance of exploring this potential avenue in what is otherwise a neglected area of research. Successful research into the effect of PAG and STN stimulation on PEFR can now be parlayed into examining patients with dyspnoea as a predominant issue. For example, exploration of DBS in patients with abnormal lower airway calibre and established chronic lung disease to confirm the benefits of controlling the lungs via the brain, and to more fully understand the mechanisms.

Although we are yet to map out the autonomic nervous system in a way that provides full mastery over it, we have enough tantalizing leads to create targets that provide invaluable relief from a whole host of distressing symptoms, regardless of incomplete mechanistic framework. Lest we not forget, this is a compromise that has worked very well for Parkinson’s and tremor patients. One of the appealing features of trialling neuromodulation in refractory states is its reversibility. This is partially because there is a sense of the ability to ‘undo’ should it not go to plan, and partially because this on–off ability may prove useful as part of the treatment. For example, for those with orthostatic hypotension, stimulation could be halted overnight when patients are supine, preventing the nocturnal hypertension these patients experience using current medications [123]. Additionally, neuromodulation offers the ability to change settings as symptoms change, compared with, for example, the ‘one shot’ opportunity of a cingulotomy for intractable cancer pain [124].

The use of neuromodulation could revolutionise symptom-control in the near future, providing services for those difficult to reach under current regimes. The challenges are to justify the initial cost of surgery, carefully select the right patients, and acquiesce neurophobia for all involved (medical team, patients, relatives, and funding bodies). We can now start to tap into the potential benefits in a manner akin to what has already developed for motor symptoms.

## Figures and Tables

**Table 1 brainsci-09-00232-t001:** Key studies involving DBS and Blood Pressure.

Author	*N*	Patient type	Target	Results	Conclusion
**Holmberg et al. 2005** [34]	19 DBS; 10 controls with no stimulation (optimally medicated)	PD ^a^	STN ^b^	After 1 year of treatment, HRV ^c^ and BP ^d^ during tilt was reduced compared to baseline; but no difference between stimulated and non-stimulated group.	No beneficial results for orthostatic hypotension.
**Green et al. 2005** [35]	15	PAG Chronic pain	PAG ^e^	In patients at rest in a seated position, stimulation using dorsally located PAG electrodes produced an elevation of approximately 16mmHg in systolic BP, whereas stimulation using ventrally located PAG electrodes caused a decrease of approximately 14mmHg in this parameter.	PAG stimulation alters BP and is dependent on whether stimulation is dorsal or ventral.
**Lipp et al. 2005** [19]	5	PD	STN	Electrodes inadvertently placed close to posterior hypothalamus showed BP and respiratory rate increased after stimulation.	PHA ^f^ stimulation may increase BP.
**Green et al. 2006** [36]	11	PAG Chronic pain	PAG	Patients experienced decreased systolic BP in ’on’ stimulation when moving from sitting to standing; in one patient with clinical orthostatic hypotension, systolic BP fell by 15% from baseline (145–148 mmHg) on changing from a sitting to standing position without stimulation, compared with a change of only 0.1% on stimulation. For those with mild orthostatic hypotension the effects were reversed. For those with orthostatic hypotension, no side effects were experienced.	Stimulation of the PAG can prevent orthostatic hypotension.
**Stemper et al. 2006** [37]	14	PD	STN	DBS STN, increased HR, decreased blood flow to skin, and maintained BP after 60 degrees HUTT.	Beneficial results for orthostatic hypotension.
**Green et al. 2007** [38]	1	chronic pain (oral cavity/soft palate pain)	PAG	Hypertensive patient with PAG stimulation for Chronic pain experienced their baseline a fall in arterial pressure.	PAG may be suitable as a HTN treatment.
**Ludwig et al. 2007** [39]	29 (14 DBS; 15 Controls	PD	STN	BP (and HR) during rest and orthostatic conditions did not differ significantly between groups.	No beneficial results of DBS for orthostatic hypotension.
**Cortelli et al. 2007** [40]	8	Cluster headache	PHA	During HUTT ^g^ systolic BP maintained when ’on’ stimulus but fell by 3% when ’off’. Ratio of low:high frequency components in HRV increased during on stimulation.	PHA stimulation could aid orthostatic hypotension. CV (including diastolic BP) changes appear to be hypothalamic-mediated sympathetic activation.
**Patel et al. 2011** [41]	1	central pain syndrome- left hemibody pain	PAG	Despite pain returning to baseline four months after surgery, DBS continued to affect BP as indicated by blood pressure rise of 18/5 mmHg.	Despite pain returning to baseline four months after surgery, DBS continued to affect BP thus affect of DBS on BP is not just relating to pain relief.
**Sverrisdottir et al. 2014** [42]	17 (7 PAG, 10 STN)	chronic pain/ PD	STN	Increase in orthostatic tolerance.	Beneficial results for orthostatic hypotension.
**O’Callaghan et al. 2017** [43]	1	chronic pain	ventral PAG	After 6 months of chronic low frequency DBS of vPAG, BP lowered from 280 to 210–230 systolic.	Possible use of PAG as therapy for intractable HTN.

^a^ PD = Parkinson’s Disease; ^b^ STN = Subthalamic nucleus; ^c^ HRV = Heart rate variability; ^d^ BP = Blood Pressure; ^e^ PAG = Periaqueductal gray; ^f^ PHA = Posterior hypothalamic area; ^g^ HUTT = Head uptilt table test.

**Table 2 brainsci-09-00232-t002:** Unpublished Registered Clinical Trials relating to DBS and Palliative symptoms.

Area	Title	Status	Conditions	Interventions	Locations
Blood Pressure	Deep Brain Stimulation for Autonomic and Gait Symptoms in Multiple System Atrophy	Recruiting	Multiple System Atrophy|Autonomic Failure|Postural Hypotension|Bladder, Neurogenic|Gait Disorders, Neurologic	Procedure: Deep brain stimulation	John Radcliffe Hospital, Oxford, Oxfordshire, United Kingdom
Chronic pain	Deep Brain Stimulation (DBS) for Chronic Neuropathic Pain	Recruiting	Chronic Neuropathic Pain|Post Stroke Pain|Phantom Limb Pain|Spinal Cord Injuries	Device: Active DBS|Device: Inactive DBS	University of California, San Francisco, California, United States
Chronic pain	Safety Study of Deep Brain Stimulation to Manage Thalamic Pain Syndrome	Completed	Chronic Pain	Device: Deep Brain Stimulation for Thalamic Pain Syndrome	Cleveland Clinic Foundation, Cleveland, Ohio, United States
Chronic pain	Combined Cingulate and Thalamic DBS for Chronic Refractory Chronic Pain	Not yet recruiting	Chronic Refractory Neuropathic Pain	Procedure: Deep brain Stimulation of cingulum anterior	Department of neurosurgery, Nice, France
Respiratory dysfunction	The Effects Of DBS Of Subthalamic Nucleus On Functionality In Patients With Parkinson’s Disease: Short-Term Results	Recruiting	Parkinson Disease|Surgery|Respiration; Decreased|Muscle Weakness	Device: Maximum Inspiratory Pressure and Maximum Expiratory Pressure	Hatay Mustafa Kemal University, Antakya, Hatay, Turkey
Urinary dysfunction	Effect of Deep Brain Stimulation on Lower Urinary Tract Function	Completed	Movement Disorder|Urinary Tract Disease	Procedure: deep brain stimulation ON|Procedure: Deep brain stimulation OFF	Department of Neurology, University of Bern, Bern, Switzerland|Department of Urology, University of Bern, Bern, Switzerland
Urinary dysfunction	Deep Brain Stimulation in Patients With LUTS	Recruiting	Bladder Dysfunction|Neurogenic Bladder	Other: Cohort	Houston Methodist Research Institute, Houston, Texas, United States
Other dysfunction	Deep Brain Stimulation and Digestive Symptomatology	Completed	Parkinson’s Disease	not specified	Rouen University Hospital, Rouen, France

**Table 3 brainsci-09-00232-t003:** Key Studies surrounding DBS and Dyspnoea.

Author	N	Patient Disease	Target	Outcome Measures	Results	Conclusion
**Hyam et al. 2012** [61]	17 chronic pain; 20 movement disorder; 7 control thalamus; 10 control Gpi	Movement disorder and chronic pain	STN ^a^ and PAG ^b^ (sensory thalamus and GPi ^c^ as control)	PEFR ^d^,FEV1 ^e^	STN and PAG improved PEFR but not FEV1; Patient with obstructive lung function showed improved FEV1 on stimulation.	Possible to control the lungs via the brain.
**Vigneri et al. 2012** [76]	6 PD; 5 cluster headache	PD ^f^ and cluster headaches	STN and PHA ^g^	RR ^h^ (and HR ^i^, BP ^j^)	No change on vs. off stimulation for any values when supine.	Failure to find effect.
**Xie et al. 2015** [77]	1	PD	STN	Respiratory dyskinesia	Stimulation relieved levodopa-induced respiratory distress even in medication ’on’ phase.	Not clear how DBS controls respiratory dyskinesia.

^a^ STN = subthalamic nucleus; ^b^ PAG = Periaqueductal gray; ^c^ GPi = globus pallidus; ^d^ PEFR = Peak expiratory flow rate; ^e^ FEV1 = Forced Expiratory Volume in 1 second; ^f^ PD = Parkinson’s disease; ^g^ PHA = posterior hypothalamic area; ^h^ RR = respiratory rate; ^i^ HR = heart rate; ^j^ BP = blood pressure.

**Table 4 brainsci-09-00232-t004:** Key Studies surrounding DBS and Micturition.

Author	Patient *N*	Patient Type- Disease And Brain Area	Target	Outcome Measure	Results	Conclusion
**Finazzi-Agro, Peppe et al. 2003** [88]	5	PD ^a^	STN ^b^ bilateral	Bladder compliance and capacity, first desire to void volume, bladder volume (reflex volume) and amplitude of detrusor hyperreflexic contractions, maximum flow, detrusor pressure at maximum flow and detrusor-sphincter coordination	Bladder capacity and reflex volume were increased for ’on’ stimulation. No significant differences in other parameters.	STN stimulation seems to be effective for decreasing detrusor hyperreflexia in PD. Hence a role for basal ganglia in voiding control.
**Seif et al. 2004** [90]	16	PD	STN	Filled bladder with isotonic saline and measured intiial desire to void, maximal bladder capcity, detursor contractions, detrusor pressure, and compliance of bladder.	Maximal bladder cacpity and volume at desire to void are both significantly higher during ’on’ stimulation.	DBS of STN normalises ’storage’ phase of micturition.
**Herzog et al. 2006** [89]	11	PD	STN bilateral	PET ^c^ studies measuring regional cerebral blood flow to ACC ^d^ during bladder filling.	During bladder filling, ‘off’ stimulation demonstrates increased regional blood flow to ACC and lateral frontal cortex.	Stimulation aids more effective integration of afferent bladder information.
**Winge and Nielson 2010** [78]	107	PD	STN	Danish Prostate Symptom Score; International prostate symptom score.	Patients with DBS had lower reporting of nocturia compared to patients with apomorphine pumps.	STN DBS results in lower levels of nocturia.
**Green, Stone et al. 2012** [91]	6 PAG; 2 control thalamus	Chronic pain	PAG ^e^	Sensation of filling and desire to void during saline infusion ’on’ and ’off’ stimulation.	On’ stimulation: Volume urine first escaped from penis was higher ’on’, and subjective bladder capacity was increased. This did not affect volumes at which voiding was desired.	It is possible PAG stimulation can switch off micturition.
**Aviles-Olmos, Foltynie et al. 2011** [102]	1	PD	PPN ^f^ rt side	NA	Detrusor overactivity/urge incontinence after DBS surgery. Voiding normal. Symptoms improved 6 months post-op with antimuscarinics.	Involvement of pontine micturition centres resulted in urge uncontinence.
**Roy et al. 2018** [108]	6 (5 complete)	PD	PPN bilateral	Bladder volume at maximal capacity (also looked at white matter tractography).	Increase in maximal bladder capacity when ’on’ stimulation.	PPN may be a target to alleviate some LUTD symptoms.
**Kessler, Burkhard et al. 2008** [92]	7	Essential Tremor	Thalamus	Bladder volume at first desire to void, maximal cystometric capacity.	Stimulation decreases bladder volume at ’first’ and ’strong’ desire to void, and maximal bladder capacity.	Thalamus may have a role in micturition but is not a target for rectifying dysfunction.

^a^ PD = Parkinson’s Disease; ^b^ STN = Subthalamic Nucleus; ^c^ PET = Positron Emission Tomography; ^d^ ACC = Anterior Cingulate Cortex; ^e^ PAG = Periqueductal Gray; ^f^ PPN = Pedunculopontine Nucleus.

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
