# Peer review of "The Use of Neuromodulation for Symptom Management"

_brainsci, 2019, doi:10.3390/brainsci9090232_

Round 1

Reviewer 1 Report

This paper reviews the evidence of the effects of neuromodulation on the symptoms of pain, dyspnea, and micturition and the management of blood pressure. The authors provide an excellent review of the pathophysiology of how neuromodulation may be affecting these symptoms. Reviewing the benefits of neuromodulation for symptom management adds important information to the existing literature. There are, however, some major edits for consideration.

Major edits:

The authors describe the use of neuromodulation for symptoms that affect quality of life. The authors use the term "palliative care" somewhat liberally. They include some symptoms and diseases that are not particularly managed by palliative care. Specifically, cluster headaches are primarily managed by neurologists and blood pressure/orthostatic hypotension/hypertension is primarily managed by primary care physicians or cardiologists. I would recommend the authors consider changing the title to "The Use of Neuromodulation for Symptom management.” Different underlying causes affect the type of pain experienced by patients. Therefore, the effect of neuromodulation on different types of pain may not be generalized and the authors should address this in their assessment.  In the Dyspnoea section, the authors state, "…are all too aware that pharmacology has neglected the distressing but difficult to treat symptom of dyspnoea" (lines 198-199). This statement refers to a paper by Ambrosino and colleagues (2019) in which what the authors describe is specific to the "most recent evidences on symptomatic pharmacological and non-pharmacological therapies for the management of dyspnoea in patients with advanced CRD either in stable state or at their end of life (EoL), independent of the underlying respiratory disease." This may not be generalizable to the COPD population as the authors are describing here. Additionally, this statement is not clear about what specifically "pharmacology has neglected" (line 198) as the authors write. The authors describe difficulty in managing dyspnea specifically for COPD. Refractory dyspnea not managed by standard care treatments is often end stage lung disease, which would include severe COPD. I recommend the authors reconsider describing the effects of neuromodulation for end stage lung disease. If these data are not available, then I recommend discussing dyspnea more specifically for the disease of COPD and not generalized dyspnea. 

Minor points:

The authors should define "SCS" on line 67.

Author Response

We thank the reviewers and editors at Brain Sciences for taking the time to read and review our article favourably.  Your comments have been most useful.  We have attempted to address all edits as per your advice.  Please see below for individual comments.

Reviewer 1

Thank you for your  kind comments and for highlighting the use of reviewing neuromodulation of symptom management.

Regarding major edits:

Liberal use of ‘palliative care’: Thank you for opening this conversation. Our discussion is based around improvement of quality of life by altering otherwise intractable symptoms.  We  agree cluster headaches and blood pressure may be better described as ‘symptom control’.  We have altered the title from ‘palliative care’ to ‘symptom management’ as per your suggestion, and broadened the abstract,  introduction  and conclusion to better reflect this.

‘Different underlying causes effect the types of pain experienced by patients. Therefore, the effect of neuromodulation on different types of pain may not be generalised and the authors should reflect this in their assessment’. Thank you for this comment, information pertaining to this had previously been edited out  (instead referring a previous review article) for brevity, with just a quick mention of ‘heterogeneity of pain aetiology’ (line 84 of track changes version of the document). We have now expanded this to explain that whilst different causes/types of pain should be treated in different ways, we do not believe it is as simple as matching neuromodulatory procedure to the pain aetiology (lines 84-90).

Treatment of dyspnoea: Ambrosino reference may not be generalisable to COPD. Thank you for pointing this out, we have removed this from line 271.  (See next point)

‘I recommend the authors reconsider describing the effects of neuromodulation for end stage lung disease. If these dates are not available, then I recommend discussing dyspnoea more specifically for COPD and not generalised dyspnoea.’  We have now altered this section to specify its relevance to COPD specifically, thank you for pointing this out.

Minor edit: Added definition of SCS to line 77

Reviewer 2 Report

This is an interesting paper and reviews some of the literature on the subject.

It would be improved with more discussion of the possible use / evidence from patients who are receiving neuromodulation, eg for Parkinson's disease, and have shown / may benefit in symptom management

There is little discussion of the use of neuromodulation at  the end of life and a short section on the ethical issues / problems would be helpful

There is great use of abbreviation and these are not always explained

eg page 2 line 63 (SCS)  should be added, so line 67 makes sense

There are some other typos / use of wording that could be improved;

Page 2 line 82 versus should be in full, not vs

Page 4 line 155 "makes sense" is cumbersome - could be: The effects of PAG stimulation on the cardiovascular system may be understood as there is evidence....

Page 5 line 209 The should be the

Page 6 line 234 "Make sense" should be changed; line 247 Asthma could be asthma

Page 10 line 380 lesser should be a reduced; line 381 these changes depend

Page 11 line 402 "Moving on to" is clumsy and could be expressed more clearly; line 413 Tremor could be tremor

Author Response

We thank the reviewers and editors at Brain Sciences for taking the time to read and review our article favourably.  Your comments have been most useful.  We have attempted to address all edits as per your advice.  Please see below for individual comments.

Reviewer 2:

We thank the reviewer for the their most helpful comments and appreciate attention to detail regarding minor comments.

Regarding Major edits:

‘It would be improved with more discussion of the possible use/evidence from patients who are receiving neuromodulation, eg for Parkinson’s disease, and have shown/may benefit in symptom management.’ Thank you for the comments. As you know there are certain conditions for which we have class 1 evidence e.g PD and dystonia, but regretfully this is not yet the case regarding the amelioration of the symptoms discussed. The patients we describe have received neuromodulation for movement disorder or chronic pain; their dysautonomias were either serendipitously relieved or have been relieved to some extent through the trials already noted. Paragraph starting line 30 hopefully sets the tone for this, with the use of PD patients previously fitted with DBS to assess blood pressure control highlighted line 251-253, line 285-286 noting the use of chronic pain patients for breathlessness studies, and 423-425 reiterating the use of PD DBS patients for micturition studies.  We hope this addresses the comments.

‘There is little discussion of the use of neuromodulation at the end of life and a short section on the ethical issues/problems would be helpful.’ Yes thank you for these suggestions. We have now expanded our title ‘palliative’ to ‘symptom control’ to better reflect the nature of the article and moved away from the notion of ‘end-of-life’ as per reviewer 1’s comments.  We agree a section on the ethics of relieving intractable symptoms may be useful and this has been added accordingly (lines 555-569). Thank you for this recommendation.

Minor edits: All comments pertaining to minor corrections have been corrected as per your suggestion, thank you for noting these.

Round 2

Reviewer 1 Report

The authors have appropriately addressed all of my previous questions and comments. This paper is acceptable for publication.